# Computational Analysis of Exotic Molecular and Atomic Vibrations in Ice XV

**DOI:** 10.3390/molecules24173115

**Published:** 2019-08-27

**Authors:** Xiao-Ling Qin, Xu-Liang Zhu, Jing-Wen Cao, Lu Jiang, Yue Gu, Xue-Chun Wang, Peng Zhang

**Affiliations:** School of Space Science and Physics, Shandong University, Weihai 264209, China

**Keywords:** ice XV, translational modes, hydrogen bond, CASTEP, density functional theory

## Abstract

It is always difficult to assign the peaks of a vibrational spectrum in the far-infrared region. The two distinct peaks seen in many ice phases are still a mystery to date. The normal modes of ice XV were calculated using the CASTEP code based on first-principles density functional theory. On the basis of vibrational modes analysis, we divided the translational modes into three categories: four-bond vibrations, which have the highest energy levels; two-bond vibrations, which have medium levels of energy; and relative vibrations between two sublattices, which have the lowest energy. Whale et al. found that some intramolecular stretching modes include the isolated vibration of only one O–H bond, whereas the others do not vibrate in ice XV. We verified this phenomenon in this study and attributed it to local tetrahedral deformation. Analysis of normal modes, especially in the translation and stretching band of ice XV, clarified the physical insights of the vibrational spectrum and can be used with other ice phases.

## 1. Introduction

More than 17 ice phases have been experimentally identified at various temperatures and pressures [1]. These phases can be divided into two main categories on the basis of hydrogen long-range order or disorder. Among ice phases, seven hydrogen-ordered phases are currently known, ices II, VIII, IX, XI, XII, XIV, and XV. Most of them can be converted into hydrogen-disordered ice phase, in which the water molecules exhibit orientation disorder after heating. There are six known counterparts: Ih/XI, III/IX, V/XIII, VI/XV, VII/VIII, and XII/XIV [2]. The only exception is ice II, for which a hydrogen-disordered counterpart has not been observed. Upon heating, it transforms to either ice Ill or ice V depending on the pressure. The high-pressure phase of ice VI was first discovered by Bridgman [3]. The crystal structure of ice VI was solved by Kamb via X-ray diffraction at ambient pressure at the temperature of liquid nitrogen [4], and confirmed by Kuhs et al. via neutron diffraction [5,6]. Considering only oxygen lattices, the primitive cell of ice VI contains 10 water molecules with space group P4_2_/nmc, which is a hydrogen-disordered phase. Ice VI consists of two interpenetrating yet independent hydrogen-bonded networks, and has thus been described as a “self-clathrate” [4].

In 2009, Salzmann et al. found a new hydrogen-ordered phase by doping HCl in ice VI at a lower temperature, which is called ice XV. This hydrogen-ordered structure is antiferroelectric $\left(P\bar{1}\right)$ and can be stable from 0.8 GPa to 1.5 GPa at temperatures below 130 K [7]. Whale et al. measured the Raman spectrum and performed simulations with the first-principles density functional theory (DFT) [8]. Salzmann et al. later presented another hydrogen-ordered structure of VI, the ferromagnetic structure with space group Cc [9,10]. In 2018, Gasser et al. demonstrated the existence of one more XV phase, which they called the ice beta-XV phase [11].

For theoretical investigations of vibrational modes, the most interesting results were those of Whale et al., who found exotic “rigid network” molecular vibrations in the translation band and “isolated stretching” modes of atomic vibrations in the stretching band [8]. We also found “rigid network” modes in ice VIII and VII [2] and “isolated stretching” modes in ice XIV [12]. According to our series of investigations of ice phases, we identified two intrinsic categories of molecule vibrations in the translational band that constitute two sharp peaks [2,12,13,14,15]. In this paper, we confirm the presence of cluster vibrations in a much lower region of the translation band. We review the normal modes in ice XV and aim to clarify the physical insights of these exotic vibration modes.

## 2. Computational Methods

We first constructed the oxygen structure of ice VI. Then, we used a self-compiled program to find two possible hydrogen-ordered structures. We selected the antiferromagnetic one that is the same as reported by Whale for this work. The density is 1.32 g/cm^3^, in which the three orthorhombic constants of a primitive cell are a = 6.23 Å, b = 6.22 Å, and c = 5.83 Å. Using the CASTEP [16] code based on first-principles density functional theory (DFT), we calculated the phonon density of states (PDOS) of antiferroelectric ice XV. The energy threshold and self-consistent field tolerance were set as 1 × 10^−9^ eV/atom. The energy cutoff was 750 eV, and the k-point was 2 × 2 × 2. As the electron density has a large distribution in ice lattice, we chose generalized gradient approximation (GGA) for this work. To test the exchange-correlation (XC) functional, we tried RPBE, PBE, PW91, WC, and PBESOL separately for geometric optimization. The results showed that the densities are 1.21, 1.40, 1.41, 1.55, and 1.56 g/cm^3^, respectively. As the length of intermolecular hydrogen bond (H-bond) is relative to volume change and the vibrational frequency is very sensitive to H-bond length, the best selection is RPBE [17], which produces a slight redshift in the translation band, but greater accuracy in the intramolecular vibrations. The hydrostatic pressure was set to 0.9 and 1.4 GPa for comparison with the k mesh was set as 4 × 4 × 4. We found the H-bond frequencies of 1.4 GPa have a big redshift than 0.9 GPa (please see Appendix A). Thus, we selected the simulation data of 0.9 GPa for discussions hereinafter. The optimized dimensions are a = 6.39 Å, b = 6.44 Å, and c = 6.02 Å.

## 3. Results and Discussion

Figure 1 shows the computing spectra of Raman scattering, infrared (IR) absorption, and PDOS, which are divided into four parts according to four separate vibrational regions. Because of the wide scale of the intensity in different regions, we adjusted their proportions for comparison. A primitive cell contains 10 molecules; thus, there are 10 × 3 × 3 − 3 = 87 optical normal modes.

Table 1 shows the details of PDOS, normal modes, and Raman scattering peaks. Column 1 lists the main peaks of PDOS. We identified 34 distinguishable peaks, of which 11 are major. The PDOS curve integrates phonon signals throughout the Brillouin zone (BZ), so it could correspond to an inelastic neutron scattering (INS) spectrum. Because reports of INS data are lacking, we present PDOS herein for reference. The 87 normal modes are shown in column 2, and the IR and Raman active intensities are shown in the next two columns. Sixteen Raman peaks are measured in the experiment: 114, 158, 186, 220, 230, 300, 312, 387, 451, 499, 548, 3224, 3335, 3345, 3396, and 3425 cm^−1^ [8]. However, the peak at 387 cm^−1^ did not match any normal mode. We doubt that this is a false peak because a gap should be present between the translation and libration bands. There is a slight underestimation of the hydrogen bond (H-bond) by RPBE, but it is more accurate than Whale’s work by PBE [8].

In the translational band, the Raman peaks are from optic modes near the BZ that are detectable subject to selection rules. On the basis of harmonic approximation, 27 optic normal modes are present between 73 and 285 cm^−1^. Note that the peaks are matched with normal modes by the calculated intensities.

Our previous study of ice Ic revealed two kinds of intrinsic translational modes in the ice lattice [13]. In the strong mode, the molecule vibrates together with four connecting H-bonds, which is called the four-bond mode. Two perpendicular degenerate weak modes involve only two oscillating H-bonds, which is called the two-bond mode. Treating the two H-bonded molecules as spring, a simple harmonic oscillator model yielded a strength ratio of $\sqrt{2}$.

Although the tetrahedral structure of ice XV presents deformation under pressure to some extent, these two kinds of modes can also be distinguished in the translation region. For the mode at 285 cm^−1^ shown in Figure 2, typical molecules vibrate along its HOH angular bisector, which yields a strong vibrational mode (please see Appendix A). Note that some small vibrations of other molecules can keep the mass center of a primitive cell static. The mode at 206 cm^−1^ is a two-bond mode, in which a molecule vibrates toward its two neighbors (please see Appendix A). Whale et al. found that the three lowest modes at 66, 78, and 85 cm^−1^ present vibrations of the two rigid ice XV networks. In this study, the three modes are at 73, 106, and 107 cm^−1^, respectively. They actually represent three orthorhombic vibrations in 3D space that originate from the interactions of two sublattices without H-bonds, and thus present the lowest vibrational energy. Figure 2 illustrates an example of the mode at 107 cm^−1^ (please see the dynamic process from Appendix A). We also found this phenomenon in ice VII and VIII [1]; however, the most common case in other ice phases is that many cluster vibration or skeleton deformation modes are present in the much lower region. Considering the translational band of ice, we can divide normal modes into three categories: four-bond vibrations, which possess the highest energy; two-bond vibrations, with medium energy levels; and cluster vibrations, which have the lowest energy. Figure 3 shows a diagram of fitted curves of three kinds of modes in ice XV compared with the inserted PDOS. Note that the data for normal modes are in the BZ center, whereas the real PDOS covers the entire first BZ. Thus, some disagreements occur as a result of the dispersion of phonons.

As shown in Figure 2, the libration band contains 30 normal modes from 477 to 936 cm^−1^. There are three types of vibration modes: rocking, or rotation of the whole molecule around an axis perpendicular to the molecular plane; twisting, or rotation around an axis coincident with the HOH angle bisector; and wagging, or rotation of the molecule around an axis in the molecular plane, perpendicular to the bisector of the HOH angle. For an optic mode, the mass center of the primitive cell remains static while the molecules within vibrate. Figure 2 shows that the main vibrating mode at 477 cm^−1^ is rocking, whereas the mode at 913 cm^−1^ is twisting. For the highest frequency at 936 cm^−1^, the vibrating mode of molecules is a mixture of wagging, rocking, and twisting.

The bending band has 10 normal modes from 1609 to 1709 cm^−1^, as shown in Figure 4. The vibration pattern of each molecule is bending, whereas the collective vibrations can be distinguished as in-phase and out-of-phase. For the highest-frequency mode at 1709 cm^−1^, the vibrating mode of crystal lattice is in-phase. The lowest mode at 1609 cm^−1^ is out-of-phase. The mode at 1684 cm^−1^ shown in Figure 4 has the greatest intensity of IR absorption, but has no Raman activity. In contrast, the mode at 1655 cm^−1^ has the greatest intensity of Raman activity, but no IR activity in this band. Because the intensities are very small compared with the stretching band, it is always difficult to detect signals for IR and Raman experiments.

There are 20 normal modes in the stretching band from 3278 to 3485 cm^−1^, including intramolecular symmetrical and asymmetrical stretching vibrations. For the lowest frequency at 3281 cm^−1^, all molecules show symmetric stretching. For the frequency at 3461 cm^−1^, the main vibrations show asymmetric stretching. The mode at 3280 cm^−1^ has the greatest intensity of Raman activity, but no IR activity, whereas the opposite is found in the mode at 3278 cm^−1^.

Whale et al. reported the normal modes in this band; some vibrational modes include isolated vibration of only one O–H bond, whereas the other does not vibrate [8]. A typical example is shown in Figure 4 of the mode at 3367 cm^−1^ (please see Appendix A). We also found this vibration pattern in other ice phases, such as ice XIV and VII. We regard this “isolated stretching” mode as a special case of asymmetric stretching and attribute it to local tetrahedral deformation by pressure and a hydrogen-disordered lattice.

## 4. Conclusions

It is traditionally difficult to assign the peaks of a vibrational spectrum in the fingerprint region. We analyze the normal modes of ice XV via DFT simulation. On the basis of our series of investigations in the translational band of ice, we divide normal modes into three categories: four-bond vibrations that possess the highest energies; two-bond vibrations that possess medium energies; and cluster vibrations that have the lowest energy. In ice XV, VII, and VIII, the cluster modes are relative vibrations between two sublattices. Thus, we can analyze the translational modes in the far IR region of ice. This study presents physical insights into the vibrational spectrum and can be used with other ice phases.

In addition, we reviewed the “isolated stretching” mode first reported by Whale et al. As we also found this phenomenon in other ice phases, we do not regard this mode as a third type in addition to symmetric and asymmetric stretching, and instead attribute it to a special asymmetric stretching due to local tetrahedral deformation by pressure and a hydrogen-disordered lattice.

## Figures and Tables

**Figure 1 molecules-24-03115-f001:**
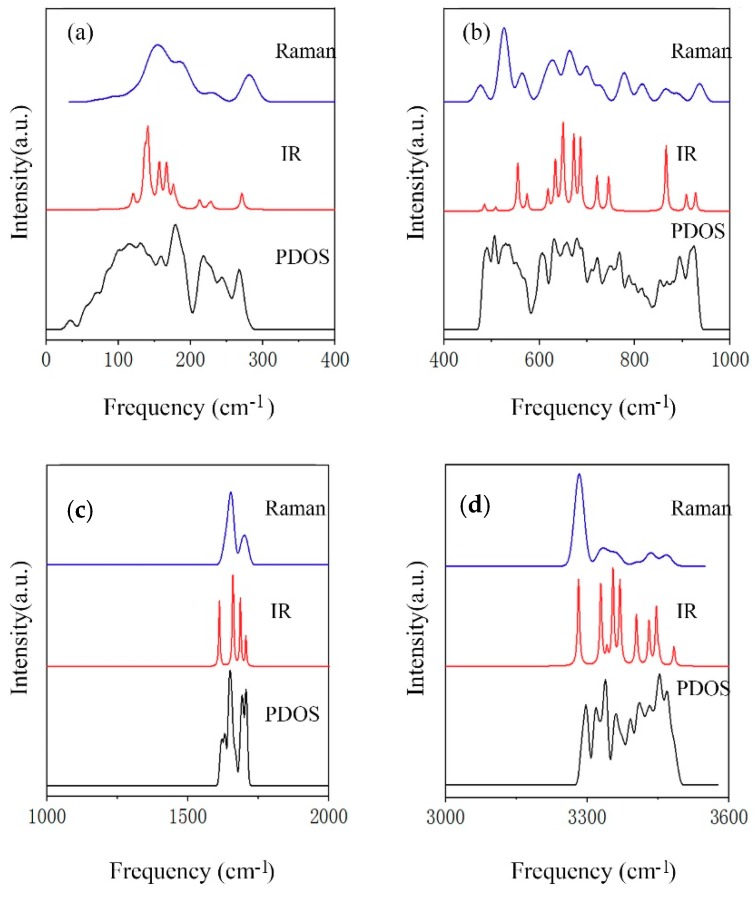
Computational spectra of Raman, IR, and phonon density of states (PDOS) of ice XV. Four vibrational bands: (**a**) intermolecular translation band; (**b**) intermolecular libration band; (**c**) intramolecular bending band; and (**d**) intramolecular stretching band.

**Figure 2 molecules-24-03115-f002:**
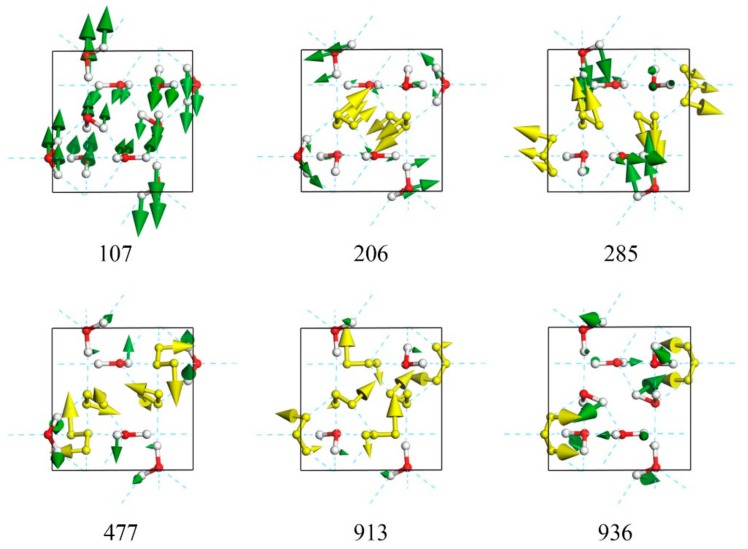
Top view of six normal modes in translational band (107, 206, and 285 cm^−1^) and libration band (477, 913, and 936 cm^−1^). Typical molecular vibrations are shown in gold.

**Figure 3 molecules-24-03115-f003:**
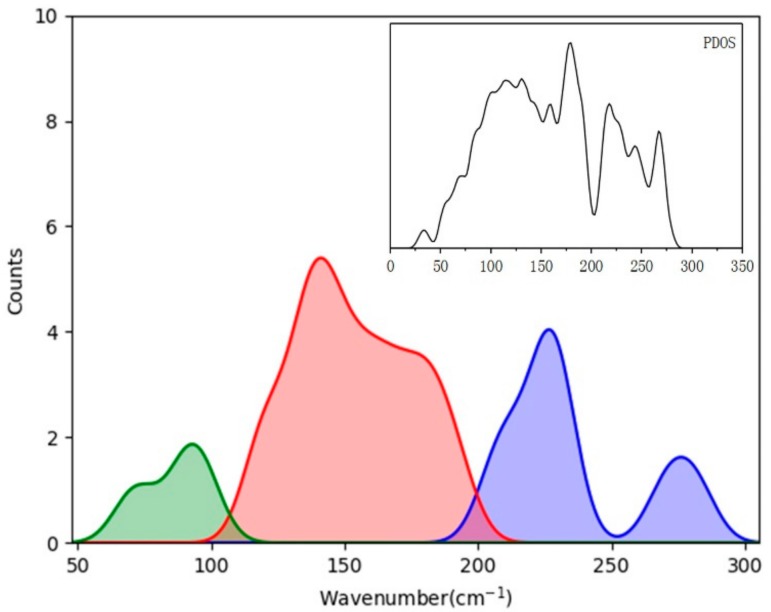
Fitted curves of four-bond modes (blue), two-bond modes (red), and non-H-bond modes (green) of ice XV in translational band. Inset PDOS curve is shown for comparison.

**Figure 4 molecules-24-03115-f004:**
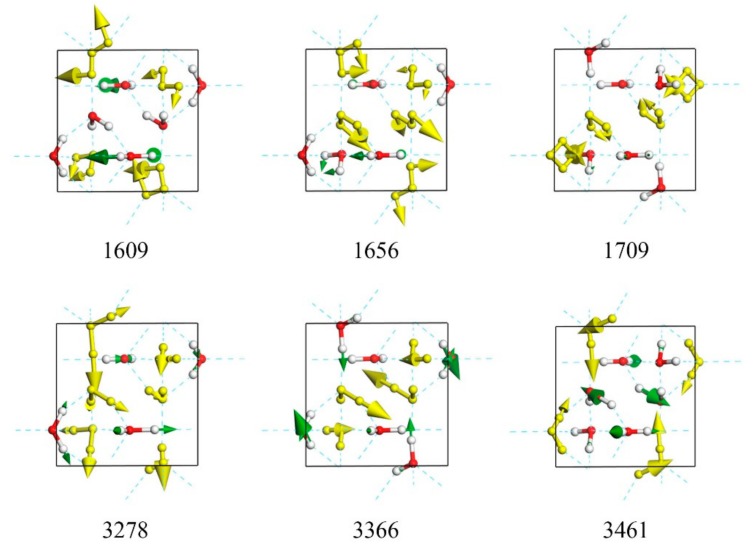
Top view of six normal modes in bending band (1609, 1656, and 1709 cm^−1^) and stretching band (3278, 3366, and 3461 cm^−1^), respectively. Mode at 3366 cm^−1^ is a typical isolated stretching vibration.

**Table 1 molecules-24-03115-t001:** Comparison of calculated results with Raman data. The normal modes (0.9 GPa) with IR/Raman intensities are compared with experimental Raman peaks. N means Raman inactive. PDOS, phonon density of states.

PDOS	Normal Modes	IR Intensity	Raman Intensity	Raman Exp. [8/11]
	73	N	0.12	
	106	N	0.24	
	107	N	0.15	
**113**	136	N	0.22	114/113
	138	1.65	N	
**130**	148	N	0.72	
	150	1.14	N	
	158	1.60	N	
	162	N	0.27	
	165	N	1.20	
	167	1.39	N	
	175	0.92	N	158/159
**159**	181	0.24	N	
	182	N	0.57	
	184	N	0.40	
**178**	185	1.22	N	
	192	N	1.55	
	201	0.22	N	186/185
	206	N	0.33	
	221	N	0.32	220/215
**217**	225	0.37	N	
	231	0.33	N	
	232	N	0.04	230/
	234	0.13	N	
**243**	236	N	0.30	300/
**268**	275	0.64	N	
	285	N	1.01	312/303
	477	N	1.81	451/450
**489**	484	1.09	N	
**506**	508	0.80	N	
**529**	531	N	7.56	499/495
	550	9.21	N	
	566	N	3.06	548/544
	579	2.93	N	
**606**	611	N	1.43	
	618	4.52	N	
	621	N	2.42	
**631**	631	10.29	N	
	635	N	3.31	
	649	1.36	N	
	652	16.64	N	
**657**	662	N	4.75	
	671	15.25	N	
**679**	677	N	1.89	
	685	8.19	N	
	702	N	3.94	
**722**	718	5.70	N	
	731	N	1.79	
**747**	748	5.20	N	
**768**	775	N	3.73	
**815**	817	N	2.33	
	864	11.87	N	
	867	N	1.51	
**894**	892	N	1.08	
	913	2.73	N	
**926**	928	3.25	N	
	936	N	1.97	
**1621**	1609	5.37	N	
**1630**	1629	N	1.94	
**1649**	1648	N	3.68	
	1655	N	5.27	
	1656	5.46	N	
	1660	3.70	N	
	1684	5.57	N	
**1692**	1692	N	2.71	
	1705	2.37	N	
**1706**	1709	N	2.91	
	3278	63.85	N	
**3296**	3280	N	3777.67	3224/3222
	3325	57.33	N	
**3318**	3326	N	552.40	3335/
**3338**	3339	7.64	N	
	3341	N	346.17	3345/3349
	3354	57.53	N	
**3360**	3357	N	288.67	
	3366	N	278.88	
	3367	50.96	N	
**3391**	3402	27.87	N	
**3411**	3403	N	164.02	3396/
	3429	27.91	N	
**3432**	3432	N	394.53	3425/3420
	3442	N	154.45	
	3448	39.83	N	
**3453**	3452	9.24	N	
	3461	N	100.59	
**3469**	3472	N	383.12	
	3485	12.40	N

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
