# Peer review of "Computational Analysis of Exotic Molecular and Atomic Vibrations in Ice XV"

_molecules, 2019, doi:10.3390/molecules24173115_

Round 1

Reviewer 1 Report

The Authors have answered my questions from the previous submission. Though the article is not perfect, it is acceptable for publication.

Author Response

Many thanks for your comments.

Reviewer 2 Report

This revised manuscript contains a number of significant improvements over the original submission. The computational results are more clearly presented. Still, the authors should consider a better comparison with experimental data - as an example, I could not find any explanation why the calculated Raman spectra in Fig. 1 appear to have broader peaks than the calculated IR spectra. Overall, I recommend publication in Molecules. This is a relevant incremental contribution, worth publishing, but not a comprehensive "go-to" paper.

Author Response

Thanks. We realized that the calculated Raman spectrum is not that clear. In the revised version, we adjusted the proportions of Raman and replaced figure 1.

Reviewer 3 Report

Normal modes and the phonon density of states of ice XV were calculated using DFT approximation. The translational modes were categorized as four-bond vibrations, two-bond vibrations, and cluster (between two sublattices) vibrations. The cluster vibrations and non-categorized “isolated stretching” modes of atomic vibrations were, for unclear reasons, called exotic. The aim of the study was “to clarify the physical insights of these exotic vibration modes”.

Personally, I have not found new physical insights in the paper. Instead, the authors report a list of calculated data and made a few superficial conclusions. The fuzzy statement of the research aim and the limited amount of obtained results make the article uninteresting for a wide audience. However, the work is done and the results can be useful for experts in the field.

Therefore, I recommend to accept the manuscript for publication in the presented form.

Author Response

Thanks, this manuscript try to explain the vibrational spectrum by modes analysis method. We focused on the two distinct peaks in the translational band and have published some investigations of many ice phases. In this manuscript, we verified the origin of two kinds of molecular modes as previous studies. Furthermore, we find the much lower peaks are from cluster vibrations (relative vibrations between two sublattices in this case). This is a complementary explanation on the translation band of ice. We inserted two sentences in the abstract as below: It is always difficult to assign the peaks of a vibrational spectrum in the far-infrared region. The two distinct peaks seen in many ice phases is still a mystery to date.

This manuscript is a resubmission of an earlier submission. The following is a list of the peer review reports and author responses from that submission.

Round 1

Reviewer 1 Report

Dear Authors,

I cannot recommend the manuscript for publishing because of very poor investigation material. The manuscript touches upon the calculations of ice only of one, single phase – and it is really not enough. Even though the paper is written and presented in a satisfactory form, it presents no worthy investigation of any topical phenomenon and copies the papers already published by these authors.

Provided the authors presented the calculations for ice in all its phases and conducted the analysis of the calculated spectra making a comparison with the experimental ones, it would cast no doubt in the sense and importance of the manuscript.

When it comes to INS studies (the authors state they have not been accomplished ever), such studies are presented in the book by Mitchell, Parker, Ramirez-Guesta, Tomkinson and stored in the ISIS base (spectra). The same is true for IR studies of ice – they are rather numerous (the book by Y. Marechal) and they could be compared with the obtained results.

Best wishes

Author Response

1) I cannot recommend the manuscript for publishing because of very poor investigation material. The manuscript touches upon the calculations of ice only of one, single phase – and it is really not enough. Even though the paper is written and presented in a satisfactory form, it presents no worthy investigation of any topical phenomenon and copies the papers already published by these authors.

Response:

As the figure shown (please see the PDF version), it was reported that there are two main peaks in the translational band of ice phases (Li, J. C.; Inelastic neutron scattering studies of hydrogen bonding in ices, J. Chem. Phys., 1996, 105, 6733-6755). The origin of these two peaks is still an unsolved question up to now. A clue was found from our study of ice Ic. To verify this is a rule among ice family, we need to investigate the ice phase individually.

2) Provided the authors presented the calculations for ice in all its phases and conducted the analysis of the calculated spectra making a comparison with the experimental ones, it would cast no doubt in the sense and importance of the manuscript.

Response:

There are more than 17 ice phases under different temperature and pressure. Most of them are structural deformation compared with ice Ih. However, two sub-lattices compose ice XV. Thus, the modes analysis are quite different. We cited the previous discovery from ice Ic. The main points are that there are two kinds of H-bond vibration modes in the translational band. We deduced this is a general rule among ice family and we verified this finding in ice XV. Furthermore, we discussed the non-H-bond vibration modes in the translational band and found that there are much lower strength modes in this region composed by inter-sublattices vibrations. Thus, we explained the third peak in the far infrared region of ice and presented new physical insights.

3) When it comes to INS studies (the authors state they have not been accomplished ever), such studies are presented in the book by Mitchell, Parker, Ramirez-Guesta, Tomkinson and stored in the ISIS base (spectra). The same is true for IR studies of ice – they are rather numerous (the book by Y. Marechal) and they could be compared with the obtained results.

Response:

Thanks. However, we cannot find INS data from database of ISIS, Rutherford-Appleton laboratory as well as IR experiments. Since this ice structure was found in 2009, the relative experiments are rare due to extreme experimental conditions until now.

Reviewer 2 Report

This manuscript presents an in-depth computational study of one of the solid phases of ice. The results of the calculations are well presented, in particular the separation into four frequency ranges. It would be an improvement if the authors could provide a more detailed comparison with experimental results for any of the four frequency regions.

The manuscript presents an interesting overview of the computational spectroscopy of hydrogen-bonded molecules and is well suited for the journal.

Author Response

1) This manuscript presents an in-depth computational study of one of the solid phases of ice. The results of the calculations are well presented, in particular the separation into four frequency ranges. It would be an improvement if the authors could provide a more detailed comparison with experimental results for any of the four frequency regions.

Response:

Thanks. We cannot find inelastic neutron scattering data as well as IR experiments. Since this ice structure was found in 2009, the relative experiments are rare due to extreme experimental conditions. Only Raman experiments are available up to now. We present simulated INS and IR results for refecence.

2) The manuscript presents an interesting overview of the computational spectroscopy of hydrogen-bonded molecules and is well suited for the journal.

Response:

Many thanks.

Reviewer 3 Report

The submitted manuscript is devoted to the theoretical study of ice polymorph. While it is a nice paper and deserves to be published, it has some drawbacks which I present below.

Line 22: "counterparts" - what do you mean? On what basis there are counterparts? Structure? Symmetry? Energy?

Lines 45-50: Where was the crystal structure of ice XV taken from? A database? What was its refcode? Was it optimized? If yes, was the cell optimized as well?

Why you chose RPBE? It was found to be very inaccurate in the other works. Why not i.e. PBESOL? I would like you to perform the geometry optimization (with unit cell dimensions as well) using different functionals and DFT-D dispersion corrections to ensure that the RPBE is the right choice.

Hydrostatic pressure of 0.9 GPa. OK, but why this value? It would be nice to repeat those calculations (PDOS) at i.e. 1.4. GPa and compare the results.

K-point 2x2x2, is it enough? What were the cell dimensions? Since the primitive cell is rather small, shouldn't it be 4x4x4?

Line 62: There is a missing space here needed.

I assume that the calculations were performed for a primitive cell. Of course, due to the plane wave code CASTEP includes periodicity, however when the unit cell is small it is encouraged to perform the calculations for the i.e. 2x2x2 or even 3x3x3 supercell.

I couldn't access supplementary materials. Where are they?

Round 2

Reviewer 1 Report

Dear Authors and Editors,

This manuscript presents very little research results. One calculation of one phase of ice! All phases of ice are so well researched and presented in the literature. My opinion is that the manuscript does not correspond to the level (impact factor 3!) of the Molecules.

Best regards

Reviewer 3 Report

The Authors have tried to answer my questions, yet I still find the manuscript not suitable for publication. Please polish your article more.

The information provided by the Authors in their response:

"Among ice phases, seven hydrogen-ordered phases are currently known, ices II, VIII,
IX, XI, XIII, XIV, and XV. Upon heating, these polymorphs transform to their
hydrogen-disordered counterparts in which the hydrogen-bonded water molecules
exhibit orientational disorder. The only exception is ice II for which a hydrogendisordered
counterpart has not been observed. Upon heating, it transforms, depending
on the pressure, to either ice III or ice V."

should be provided in the introduction as it is not a common knowledge what counterparts of ice mean.

Manuscript still lacks the information about the geometry optimization, was it performed or not? What type? Only H or O and H? What about cell dimensions? Where they optimized? How? Why or why not?

I am still not convinced about RPBE. Prior to the main calculations (PDOS) Authors should test at least few XC functionals and compare the obtained results with the experimental data. Not only density but also other experimental data. Those calculations and their discussion should be presented in the manuscript.

The results of calculations at 1.4 GPa, presented in the answer to my questions, should be provided in the article or at least SI.

In my opinion the phonon calculations using 4x4x4 should be performed. After all, the studied crystal is small and the calculations will not be extensively long.

I still don't know the dimensions of unit cell.